METHODS

# BINSEQ: A family of high-performance binary formats for nucleotide sequences

**Noam Teyssier** (ID)*, **Alexander Dobin**

Bioinformatics, Computational Tech Center, Arc Institute, Palo Alto, California, United States of America

\* noam.teyssier@pm.me

## Abstract

Modern genomics produces billions of sequencing records per run, which are typically stored as gzip-compressed FASTQ files. While this format is widely used, it is not optimal for high-throughput processing due to its reliance on single-threaded decompression and sequential parsing of irregularly sized records. This limitation is particularly problematic for applications that would benefit from parallel processing, such as read mapping, variant calling, and de novo assembly. Here, we present BINSEQ, a family of simple binary formats that enable high-throughput parallel processing of sequencing data. The BINSEQ family consists of two complementary implementations: BQ, optimized for fixed-length reads using a two-bit or four-bit encoding scheme with true random record access capability, and VBQ, designed for variable-length sequences with optional quality scores and block-based compression. We demonstrate that BINSEQ files are up to 90x faster than compressed FASTQ for parallel processing and can reduce analysis time from hours to minutes for large-scale genome and transcriptome analyses, particularly for resource-intensive applications like alignment, mapping, and de novo assembly. To facilitate adoption we provide high-performance libraries for reading and writing BINSEQ formats, native parallelization strategies with convenient APIs, and a command-line tool for conversion to and from traditional formats.

## Author summary

Modern sequencing technologies routinely generate billions of reads per experiment, yet the methods for storing and accessing these data have not kept pace. Sequencing reads remain predominantly stored in FASTQ, a text-based format designed for far smaller datasets. FASTQ's sequential parsing requirements and practical need for compression create a fundamental mismatch with modern multi-core architectures, where data access rather than computation has become the primary bottleneck. We address this problem with BINSEQ, a family of binary formats engineered for random access and native parallelization.

**Data availability statement:** All data used in this study are either computationally simulated using publicly available tools (wgsim, pbsim3) with the human reference genome GRCh38, or obtained from publicly available datasets in the NCBI Sequence Read Archive: SRR11214033, SRR36606595, and SRR37455459. All code and relevant data for reproducibility is available publicly on github: - binseq: Library for BINSEQ I/O, https://github.com/arcinstitute/binseq - bitnuc: Library for SIMD two-bit and four-bit operations, https://github.com/noamteyssier/bitnuc - binseq-bindings: Multi-language bindings for BINSEQ, https://github.com/arcinstitute/binseq-bindings - bsb: Parsing implementations for benchmarking, https://github.com/noamteyssier/binseq_benchmark - kmer_count: Naive parallel k-mer counting and benchmarking, https://github.com/arcinstitute/kmer-count - mmr: Minimap2 bindings with BINSEQ input support, https://github.com/arcinstitute/mmr - STAR: STAR branch with BINSEQ input, https://github.com/alexdobin/STAR/tree/binseq.

**Funding:** The author(s) received no specific funding for this work.

**Competing interests:** The authors have declared that no competing interests exist.

Systematic benchmarking across applications of varying computational complexity demonstrates that BINSEQ achieves 90-fold improvements in data access and maintains substantial advantages in compute-intensive tasks such as genome alignment, reducing runtimes from hours to minutes. We present two complementary implementations: BQ, optimized for simplicity and maximal throughput, and VBQ, designed for flexibility while maintaining high performance. By reconsidering the relationship between storage architecture and parallel processing capabilities, BINSEQ provides a practical solution to a critical infrastructure challenge in high-throughput genomics.

## 1 Introduction

Modern genomics routinely generates billions of sequencing records per experimental run, with data predominantly stored in gzip-compressed FASTQ [7]. While this format has remained the *de facto* standard for sequencing data storage due to its simplicity and widespread tool support, its fundamental design choices present significant operational constraints. The format's minimal specification, while facilitating broad adoption, introduces implementation variability and edge cases that compromise both compatibility and performance. A central limitation stems from its variable-length record structure, which necessitates sequential parsing even when the underlying nucleotide sequences are of fixed length. This architectural constraint, coupled with the practical requirement for data compression, creates an inherent tension between storage efficiency and processing speed.

These limitations become particularly acute in computational workflows that would naturally benefit from parallel processing, such as alignment [8,16], pseudo-alignment [5,20], and de novo assembly [2,14]. These are embarrassingly parallel operations, and the sequential nature of FASTQ parsing and decompression creates an unnecessary processing bottleneck that prevents effective utilization of modern multi-core architectures [13]. This inefficiency is most pronounced in, but not limited to, applications with low per-record computational complexity where I/O operations within decompression and parsing dominate the total execution time. The fundamental limitation of FASTQ, and other sequentially parsed formats, is that they are I/O-bound rather than CPU-bound. Consequently, application performance becomes bounded by storage medium throughput rather than available computational resources.

This misalignment between modern sequencing characteristics and legacy storage formats presents a clear opportunity for optimization. There is no standard for unaligned sequencing data and researchers have used many formats such as unaligned-SAM [17], BAM [17], and CRAM [6] – though the *de facto* standard is GZIP-compressed FASTQ [7]. There have also been many novel compression algorithms [1,3,18] and storage formats such as the Nucleotide Archive Format (NAF) [11] proposed as alternatives to general purpose compression algorithms. However, many of these approaches focus on efficient compression and decompression without

specifically considering accession properties of the underlying data which remains the limiting factor to computational efficiency.

Here, we present the BINSEQ family of formats, specifically engineered to exploit the properties of contemporary sequencing data and built for high-throughput parallel access. We introduce two complementary implementations: BQ, optimized for fixed-length reads, and VBQ, designed for variable-length sequences while maintaining high-performance characteristics. Unlike FASTQ, both formats natively support single- and paired-end reads unambiguously, eliminating the need to keep multiple synchronized files.

BQ introduces two key innovations in sequence data storage. First, it enforces fixed-size records for all sequences, enabling deterministic random access to any record without sequential parsing. Second, it employs a two-bit or four-bit encoding scheme for nucleotide representation, achieving a four-fold or two-fold improvement in space efficiency compared to the ASCII-based representation used in plain-text formats. This combination provides inherent compression without requiring additional compression algorithms, while the fixed-size record structure enables true parallel processing. Furthermore, by eliminating the storage of quality scores and sequence identifiers – elements frequently ignored by modern bioinformatics tools – BQ achieves additional space and processing efficiencies.

VBQ extends these core principles to accommodate variable-length sequences and quality score information when needed. Rather than enforcing fixed-size records for the entire dataset, VBQ organizes data into fixed-length and independent record blocks. Within these record blocks, sequences can be variable length and optionally include quality scores. Each block is optionally ZSTD compressed, and the format supports indexing for parallel access to record blocks. Although VBQ sacrifices direct record random access, it maintains exceptionally high performance for parallel processing and offers greater flexibility at the record level than BQ.

In this paper, we present a comprehensive description of the BINSEQ family designs and implementations, accompanied by extensive performance evaluations across bioinformatics applications of varying complexities. Our results demonstrate significant improvements in both processing speed and storage efficiency, establishing BQ and VBQ as compelling alternatives to traditional sequence storage formats for modern genomics workflows.

## 2 Methods

### 2.1 File format specification

**2.1.1 BQ.** BQ is implemented as a binary file format (.bq) specifically engineered for the efficient storage and retrieval of fixed-length DNA sequences (Fig 1a).

The format employs a two-bit or four-bit nucleotide encoding scheme (S1 Table) and maintains a consistent record structure to enable direct random access. Each BQ file consists of two primary components: a fixed-size header (S2 Table) and a data section containing sequence records.

The data section follows immediately after the header and consists of fixed-size records arranged sequentially. Each record comprises a flag field and sequence data, with the record size determined by the sequence length. The flag field, implemented as a 64-bit unsigned integer, precedes the sequence data and can accommodate implementation-specific metadata, filtering criteria, or quality metrics.

Sequence data storage employs a dense two-bit or four-bit encoding scheme defined by a simple lookup table (S1 Table). Invalid nucleotides are controlled via a configurable policy, allowing users to skip sequences with invalid nucleotides, replace invalids with random nucleotides, or replace them with a predetermined nucleotide. However, N nucleotides can be encoded using the four-bit encoding scheme. All nucleotide encodings are represented using little-endian byte order, with subsequences packed into 64-bit integers. If a sequence length is not a multiple of 32 or 16, the final integer is padded with zeros to maintain alignment.

There are two variants of the sequence data storage: primary and extended. The primary sequence data is meant to represent the first read pair or single-end reads, while the extended sequence data is used for the second read pair in

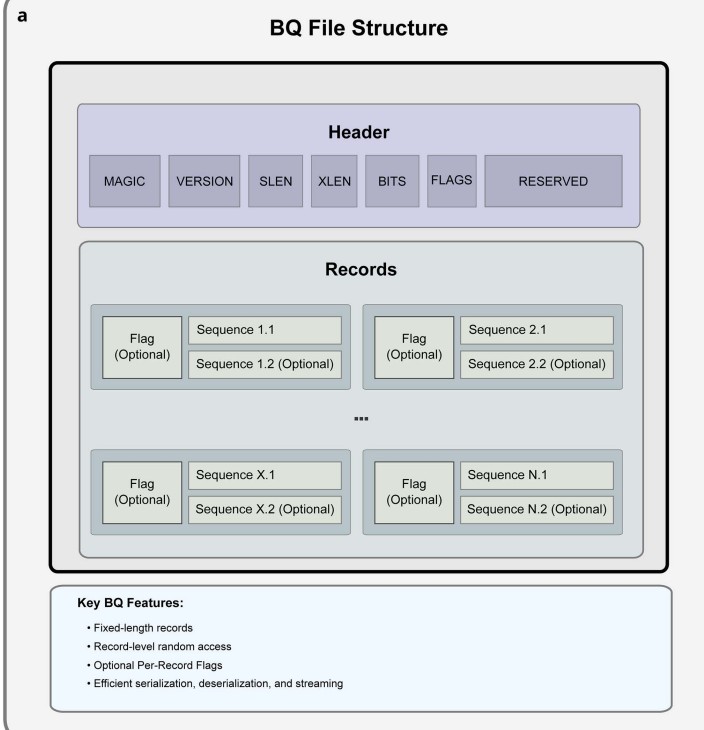

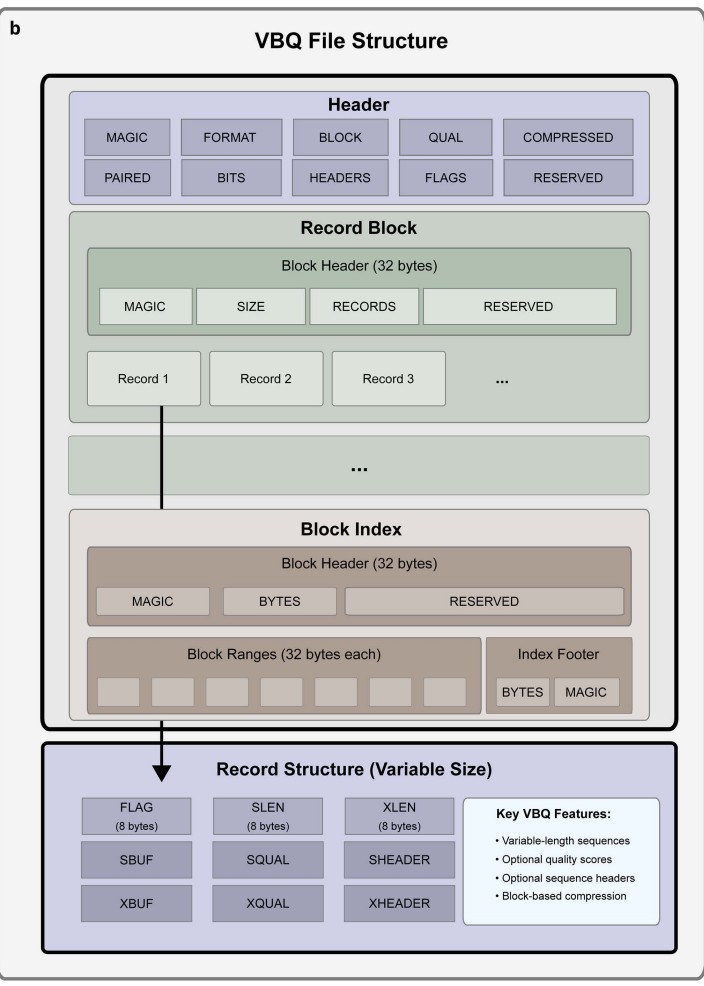

**Fig 1. BINSEQ file format family. (a)** The BQ file format. The header provides sequence-level descriptors (S2 Table). Each record is fixed length and is composed of a flag section and a sequence section. Records are optionally paired sequences and sequence lengths are described in the header. **(b)** The VBQ file format. The header provides which optional features are present in the file (S3 Table). Each record can be variably sized, include quality scores, include sequence headers, and be paired.

paired-end sequencing. The extended sequence is optional and can be omitted if not required (and is determined by the *xlen* field in the header).

For a sequence of length *N*, the total record size $S_r$ is given by:

$$S_r = F + W\left(\left\lceil \frac{N}{B} \right\rceil\right)$$

(1)

For an extended sequence pair with lengths *N* and *M*, the total record size $S_p$ is given by:

$$S_p = F + W\left(\left\lceil \frac{N}{B} \right\rceil + \left\lceil \frac{M}{B} \right\rceil\right)$$

(2)

where:

- $F$ is the flag field size (0 or 8 bytes depending on whether the FLAG is set to true in the header)

- $N$ is the sequence length in nucleotides

- $W$ is the word size (8 bytes)

- $B$ is the bits per word (e.g., 16 for four-bit and 32 for two-bit encodings)

- $\lceil x \rceil$ denotes the ceiling function

  The position $P$ of any record $i$ in the file can be calculated as:

$$P_i = H + S(i) \tag{3}$$

where:

- $H$ is the header size (32 bytes)

- $i$ is the zero-based record index

- $S$ is a record size $\{S_r, S_p\}$ calculated in Equation 1 or Equation 2

**2.1.2 VBQ.** VBQ is implemented as a binary file format (.vbq) and is designed to increase flexibility in record storage, but still allow for high parallel processing capabilities (Fig 1b).

At its core, VBQ is an extension of BQ with two notable differences: variable length records and fixed-size record blocks. Each VBQ file consists of four primary components: a fixed-size header (S3 Table), block headers (S4 Table), records (S5 Table), and a block index.

Each block has the same virtual memory size, which is specified by the file header, and packs as many complete records as possible in the block. These blocks are then optionally ZSTD compressed, and the compressed size with the number of records in the block is stored in a preceding block header. This effectively keeps each block independent and able to be processed in parallel.

Because VBQ is optionally compressed, the block header locations cannot be known absolutely, but can be easily indexed after creation to allow for parallel decompression and access. The index is a simple ZSTD-compressed binary section that annotates the start and end locations of compressed blocks in the file as well as the cumulative number of records at that block start. It begins with an uncompressed header that designates the file format as well as a quick-check on the number of bytes in the associated VBQ file to quickly check if the files are accurately paired (S6 Table). The remaining bytes are compressed and are composed of repeating ranges (S7 Table) which describe each block in the file. It finishes with the total number of bytes in the index and with a magic number to verify the integrity of the index. This index is kept at the end of the file as it is defined after write, but is loaded first when reading to allow for parallel decompression and access.

## 2.2 Implementation

**2.2.1 Nucleotide encoding.** BINSEQ employs a highly-efficient bit encoding library for nucleotide sequences. The library, *bitnuc*, provides a simple API for encoding and decoding nucleotide sequences represented as a vector of bytes, into a vector of 64-bit integers. The encoding scheme is based on a lookup table that efficiently maps nucleotide characters into two-bit or four-bit representations and packs them into 64-bit integers with efficient bitwise operations.

The implementation heavily makes use of SIMD instructions to accelerate encoding and decoding operations and is optimized for modern architectures. These instructions are available on x86 and ARM architectures and provide significant

performance improvements over scalar implementations, which are provided as fallbacks for compatibility. The library is designed to be thread-safe and can be used in parallel processing environments without additional synchronization overhead.

**2.2.2 File I/O.** BINSEQ provides a high-performance library for reading and writing BINSEQ files. This library is implemented in the Rust programming language and leverages the language's safety guarantees and performance characteristics to provide a robust and efficient implementation. The library is designed to be thread-safe and can be used in parallel processing environments without additional synchronization overhead. We have also developed both C and C++ bindings to the Rust library for multi-language support. Notably, C/C++ bindings are available for memory-mapped file access and do not currently support streaming at the time of writing though they are planned for future releases.

The compact binary format and fixed-size record structure allows for either sequential or random access to records, providing multiple access patterns for different applications. Both formats allow for direct zero-copy memory mapping, enabling very efficient access patterns.

**2.2.3 Parallel processing.** BINSEQ provides a simple hook-based interface for parallel processing (S8 Table). This interface follows a classic map-reduce pattern, allowing users to define custom map and reduce functions that operate on individual records and aggregate results on a regular batch size. This allows for efficient asynchronous processing of records in parallel during mapping but allows users to decide when best to synchronize threads for Reduce operations.

**2.2.4 Command-line tool.** Alongside the core library, we have developed a command-line tool for converting between BINSEQ and FASTQ formats, bqtools. bqtools provides a simple interface for converting between formats, allowing users to integrate BINSEQ into existing workflows.

## 2.3 Performance evaluation

To evaluate the performance of BINSEQ, we developed 3 different benchmarking scenarios to assess the format's efficiency across a range of bioinformatics applications. These scenarios are not meant to be exhaustive but rather to provide a representative sample of the performance improvements that BINSEQ can offer. The scenarios are as follows:

**Sequence Access**: This is the simplest possible scenario, where we measure the time taken to read all nucleotide sequences from a file. This is meant to replicate the raw parsing performance of the format and provide a baseline for comparison.

**$k$-mer Counting**: This is a common low-complexity bioinformatics task where we count the number of unique $k$-mers of incoming records. We select a low $k$ to measure the performance of a low-complexity application.

**Alignment**: This is a more complex scenario where we align incoming records to a reference genome. This is meant to simulate a more computationally intensive operation on each record and provide a measure of the format's performance in a high-complexity application. This scenario is more indicative of the performance improvements that BINSEQ can offer in applications like read mapping, variant calling, and de novo assembly.

Sequence access benchmarks were performed locally on a Macbook Pro M3. $k$-mer counting and alignment benchmarks were performed on a high-performance computing cluster using a Dual Intel(R) Xeon(R) Platinum 8468 CPU with NVMe solid-state drives for I/O.

**2.3.1 Sequence access.** To evaluate sequence access performance, we measured the time taken to read and potentially decode all nucleotide sequences from a file. This scenario is meant to replicate the raw parsing performance of the format and provide a baseline for comparison. To encapsulate a fundamental trade-off of modern genomics we sought not only to evaluate the speed of the format but also its storage requirements.

To represent this trade-off quantitatively we developed a normalized composite metric, $m_i$ (Equation 4), that combines the normalized time taken to read the sequences, $\bar{t}_i$ (Equation 5), and the normalized storage requirements, $\bar{s}_i$ (Equation 6), for each format $i$.

$$m_i = \frac{\bar{t}_i + \bar{s}_i}{2} \tag{4}$$

$$\bar{t}_i = \frac{t_i - \min(T)}{\max(T) - \min(T)} \tag{5}$$

$$\bar{s}_i = \frac{s_i - \min(S)}{\max(S) - \min(S)} \tag{6}$$

Each normalized metric $\{\bar{t}, \bar{s}\}$ is a value bounded by $[0, 1]$ where 0 represents the best performance and 1 represents the worst performance of that metric across all formats tested. The composite metric $m$ is the arithmetic mean of the normalized time and storage requirements, providing a single value that represents the trade-off between the two metrics for each format. This treats both time and storage requirements as equally important factors in evaluating the performance of a format, but this metric could also be adjusted to reflect different priorities.

To attempt a fair comparison between formats we evaluated multiple formats and compression strategies (Table 1). For each format we evaluated the performance of both single-threaded and parallel parsing strategies where available.

The FASTA and FASTQ parsers were implemented using the the Rust *seq_io* library [24], which offers zero-copy implementations. The SAM, BAM, and CRAM parsers were implemented using the Rust *rust-htslib* library [12], which offers FFI bindings to the C *htslib* library [4], with parallel parsing strategies. The BQ and VBQ parsers were implemented using the Rust *binseq* library, which offer zero-copy implementations with parallel parsing strategies. The Nucleotide Archive Format (NAF) parsers were implemented by wrapping the *unnaf* binary [11] and parsing sequences from stdout.

Where relevant, the Gzip, Zstd, and Lz4 decoding implementations were provided respectively by the Rust *flate2*, *zstd*, and *lz4* libraries. All libraries used were the latest stable versions available at the time of writing, and selected in good faith to provide the best performance possible for each format and compression strategy.

For FASTA, FASTQ, and NAF, the parsing strategy was limited to a single-threaded implementation. We note that while the *seq_io* library does provide parallel processing of records, the underlying byte-stream parsing is inherently sequential.

The input format for this experiment was generated using *nucgen*, a Rust library for generating uniform random nucleotide sequences. The nucleotide sequences were generated with a uniform distribution of A, C, G, and T nucleotides. Quality scores were globally "?", simplified to a single value for each sequence. This simplification will result in better performance for existing compression formats, but does not make assumptions about the distribution of quality scores. Each

**Table 1. Sequence access formats, compressions, and processing strategies.**

| File Format | Additional Compression | Parallelism | Base Encoding |
|---|---|---|---|
| FASTA [21] | {None, Gzip, Zstd, Lz4} | No | None |
| FASTQ [7] | {None, Gzip, Zstd, Lz4} | No | None |
| SAM [17] | None | {No, Yes} | None |
| BAM [17] | None | {No, Yes} | 4-bit |
| CRAM [6] | None | {No, Yes} | 4-bit |
| NAF [11] | None | No | 4-bit |
| BQ | None | {No, Yes} | 2-bit, 4-bit |
| VBQ | None | {No, Yes} | 2-bit, 4-bit |

experiment was performed with 10 Million records. For single-end reads each record had 100 bp and for paired-end reads the primary sequence had 50 bp and the secondary had 150 bp.

**2.3.2 Real data evaluation.** To complement our simulated data benchmarks, we evaluated BQ and VBQ in lossless mode (four-bit encoding with quality scores and sequence headers preserved via the `--archive` flag) on real sequencing data spanning three common data types.

For short-read single-cell RNA sequencing, we used a paired-end dataset from the combinatorial single-cell CRISPR screen study by Replogle et al. [23] (SRR11214033), sequenced on an Illumina NovaSeq 6000.

For short-read whole-genome sequencing, we used a paired-end human dataset (SRR36606595, BioProject PRJNA1394331) sequenced on an Illumina NovaSeq X Plus with 2x150bp reads.

For long-read sequencing, we used PacBio HiFi reads from a diploid genome assembly of the human H9 embryonic stem cell line (SRR37455459, BioProject PRJNA1431686).

For each dataset, the original gzip-compressed FASTQ files were converted to VBQ format using `bqtools encode` with the `--archive` flag to preserve quality scores, sequence headers, and four-bit nucleotide encoding. Where applicable (fixed-length datasets), BQ files were also generated using two-bit encoding.

We measured the resulting file sizes and record access times for both single-threaded and multi-threaded (32 threads) configurations using the same *bsb* parsing tool used in our synthetic benchmarks. For paired-end FASTQ datasets, file sizes reported are the sum of both read files. All measurements were performed on a Dual Intel(R) Xeon(R) Platinum 8468 CPU with NVMe solid-state drives, consistent with the k-mer counting and alignment benchmarks. Each benchmark was run 3 times with 2 warmup iterations, and mean runtimes are reported.

**2.3.3 K-mer mapping.** The k-mer mapping scenario is accomplished with a naive parallel implementation of k-mer counting. Briefly, each thread manages a thread-specific k-mer count table, implemented as a hashmap, and at regular batch intervals a global k-mer count table is locked, updated by the local table, and the local table counts are cleared. The k-mers are counted by sliding a window of size $k$ through the record sequence and updating the local thread table via an $O(1)$ update.

The input data for this experiment was generated using `wgsim` [17], using the HGChr38 Chromosome 1 as reference and with increasing fixed length sequences of 100, 200, and 300 basepairs.

**2.3.4 Alignment.** The alignment scenario is a more complex scenario where we align incoming records to a reference genome. To accomplish this, we adapted *minimap2* [15], a flexible and high-performance read mapper, to accept BINSEQ as input.

We make use of the Rust library *minimap2* which provides FFI bindings to the underlying C *minimap2* library. We then created a simple command-line tool, *mmr*, that accepts BINSEQ or FASTQ as input, performs the alignment operation using *minimap2*, and outputs the resulting PAF file.

For our FASTQ parsing implementation, we use the *paraseq* library, which provides a minimal-copy parser for FASTQ files with a focus on parallel processing. We matched the default parameters of the standard *minimap2* command-line tool and manage presets through the C interface.

It is important to note that the standard minimap2 command-line tool employs separate I/O threads for reading input and writing output, allowing decompression to proceed concurrently with alignment. Our *mmr* implementation similarly uses a dedicated reader thread that feeds records to the alignment worker pool. For the FASTQ path, *mmr* uses the *paraseq* library for parsing in the reader thread, while for BINSEQ inputs, record blocks are dispatched directly to workers for parallel processing. This design ensures that the comparison between formats reflects differences in data access efficiency rather than differences in I/O threading strategy.

We explored three different alignment scenarios, short-read whole-genome sequencing, long-read whole-genome sequencing, and long-read spliced sequencing. Notably, we excluded short-read spliced sequencing from this benchmark as *minimap2* only recently announced support for this use-case and STAR is one of the most common short-read spliced aligners used in the field.

To simulate short-read sequences we used the `wgsim` [17] tool, using the HGChr38 Chromosome 1 as reference and with increasing fixed length sequences of 100, 200, and 300 basepairs. To simulate long-read sequences we used the `pbsim3` [19] tool. For the whole-genome sequences we used the "wgs" strategy, the qshmm error model (QSHMM-RSII), GHChr38 Chromosome 1 as reference, and increased mean length sizes from 1000 to 9000 base pairs (sequence lengths are variable). For the spliced sequences we used the "trans" strategy, the qshmm error model (QSHMM-RSII), and the GHChr38 Chromosome 1 cDNA library as reference.

For the short-read sequence evaluation we evaluate FASTQ and BINSEQ. For the long-read sequence evaluations we evaluate FASTQ and VBQ, excluding BQ as the sequences are of variable length. For all evaluations, VBQ is compressed and including quality scores to capture the worst-case.

To evaluate the performance of short-read spliced sequences we modified the STAR aligner [8] to accept BQ as an input format via C-bindings. This implementation minimally modified the STAR aligner and as such it is not fully optimized to take advantage of the BQ format. However, it provides a proof of concept that BQ can be used as an input format for other non-rust-based bioinformatics tools.

For this task we realigned the short-read single-cell sequencing data from the Replogle et al. study [22] using either BINSEQ or gzip compressed FASTQ as input. Notably, each file was processed independently and does not make use of any file-level parallelism.

## 3  Results

Our comprehensive evaluation demonstrates that the BINSEQ family of formats delivers substantial performance improvements across all tested bioinformatics workflows. BQ and VBQ achieved up to 94x faster processing than compressed FASTQ while maintaining comparable or reduced storage requirements. These improvements were most pronounced in parallel processing scenarios, where BINSEQ formats continued to scale efficiently with increasing thread counts while traditional formats quickly reached performance plateaus.

### 3.1  Sequence access

BINSEQ formats significantly outperform traditional sequence formats in both processing speed and storage efficiency. As shown in Table 2, multi-threaded BQ demonstrates the fastest record throughput among all tested formats, while multi-threaded VBQ achieves the best combined score for processing speed and file size.

When compared to the de facto standard gzip-compressed FASTQ, multi-threaded BQ achieves more than 94x higher throughput for both unpaired records (Fig 2). Multi-threaded VBQ similarly excels with greater than 35x faster throughput. This dramatic performance improvement derives from BQ's fixed-size record structure and generally BINSEQ's elimination of sequential decompression bottlenecks that limit traditional formats.

Storage efficiency analysis reveals a competitive landscape among specialized formats. BQ (305.18 MB) maintains comparable file sizes to gzip-compressed FASTA (324.81 MB) and BAM (301.92 MB), while requiring less space than gzip-compressed FASTQ (347.43 MB). VBQ further reduces storage requirements (267.93 MB) to levels similar to CRAM (238.85 MB) and NAF (242.44 MB for FASTA, 242.47 MB for FASTQ). Notably, while CRAM and NAF achieve excellent compression efficiency, BQ and VBQ maintain superior decoding performance, especially in parallel processing paradigms.

The choice between two-bit and four-bit nucleotide encoding presents a clear performance-fidelity trade-off. Two-bit encoding provides maximum throughput and minimal storage, achieving the 94x speedups and 305.18 MB file sizes reported above for BQ. Four-bit encoding preserves ambiguous bases (N) at the cost of doubled storage (534.06 MB) and reduced throughput, though still maintaining substantial advantages over traditional formats - 40x faster than gzip-compressed FASTQ for parallel BQ (Table 2). For VBQ, four-bit encoding (299.61 MB) requires approximately 12% more storage than two-bit (267.93 MB) while maintaining greater than 14x speedup over FASTQ. The selection between

**Table 2. Comparison of different compression methods and their performance metrics across various file formats (single-end records).** Time is reported in seconds, size in megabytes. Score is a combined metric of normalized time and size (Equation 4). The "Bit Size" column indicates the number of bits used to represent each nucleotide in the format. The "Parallel" column indicates whether the method was performed in parallel (8 threads). The "Bandwidth" column indicates the Giga-bases per second (Gbp/s). The "Speedup" column indicates the runtime speedup over FASTQ.GZ.

| Format | Variant | Bit Size | Parallel (t=8) | Time (s) | Storage (MB) | Score | Bandwidth (Gbp/s) | Speedup |
|--------|---------|----------|----------------|----------|--------------|-------|-------------------|---------|
| BINSEQ | vbq | two | parallel | 0.0693 | 267.93 | **0.0131** | 14.439 | 35.10x |
| BINSEQ | bq | two | parallel | **0.0258** | 305.18 | 0.0170 | **38.696** | **94.07x** |
| BINSEQ | bq | two | serial | 0.1162 | 305.18 | 0.0289 | 8.608 | 20.93x |
| BINSEQ | vbq | four | parallel | 0.1626 | 299.61 | 0.0337 | 6.148 | 14.95x |
| BINSEQ | vbq | two | serial | 0.4133 | 267.93 | 0.0584 | 2.419 | 5.88x |
| BINSEQ | bq | four | parallel | 0.0600 | 534.06 | 0.0800 | 16.674 | 40.53x |
| BINSEQ | bq | four | serial | 0.3163 | 534.06 | 0.1139 | 3.162 | 7.69x |
| BINSEQ | vbq | four | serial | 1.0501 | 299.61 | 0.1505 | 0.952 | 2.32x |
| FASTA | lz4 | eight | serial | 0.9603 | 588.39 | 0.2127 | 1.041 | 2.53x |
| HTSlib | bam | four | parallel | 1.5342 | 301.92 | 0.2153 | 0.652 | 1.58x |
| FASTA | zstd | eight | serial | 1.4970 | 349.43 | 0.2227 | 0.668 | 1.62x |
| HTSlib | cram | four | parallel | 1.7692 | **238.85** | 0.2296 | 0.565 | 1.37x |
| FASTQ | naf | eight | serial | 1.8621 | 242.47 | 0.2435 | 0.537 | 1.31x |
| FASTA | naf | eight | serial | 1.8697 | 242.44 | 0.2440 | 0.535 | 1.30x |
| FASTA | uncompressed | eight | serial | 0.2580 | 1086.13 | 0.2475 | 3.876 | 9.42x |
| FASTQ | lz4 | eight | serial | 1.4537 | 599.43 | 0.2815 | 0.688 | 1.67x |
| FASTA | gzip | eight | serial | 1.9924 | 324.81 | 0.2816 | 0.502 | 1.22x |
| FASTQ | zstd | eight | serial | 1.9909 | 367.83 | 0.2928 | 0.502 | 1.22x |
| FASTQ | gzip | eight | serial | 2.4311 | 347.43 | 0.3446 | 0.411 | 1.00x |
| HTSlib | cram | four | serial | 3.1586 | 238.85 | 0.4093 | 0.317 | 0.77x |
| HTSlib | bam | four | serial | 3.8162 | 301.92 | 0.5161 | 0.262 | 0.64x |
| FASTQ | uncompressed | eight | serial | 0.5499 | 2068.41 | 0.5369 | 1.819 | 4.42x |
| HTSlib | sam | eight | parallel | 1.2661 | 2192.39 | 0.6636 | 0.790 | 1.92x |
| HTSlib | sam | eight | serial | 3.4933 | 2192.39 | 0.9575 | 0.286 | 0.70x |

encoding schemes should be guided by application requirements: two-bit encoding is optimal for high-quality fixed-length data where ambiguous bases are rare (e.g., Illumina single-cell RNA-seq), while four-bit encoding is appropriate when base ambiguity information must be preserved (e.g., low-coverage sequencing, variant analysis, or archival storage).

The performance gap between BINSEQ formats and traditional approaches becomes more pronounced in parallel processing scenarios. While traditional formats like FASTQ, BAM, and CRAM show modest throughput improvements with parallel processing, they quickly reach saturation, limiting their effective utilization of modern multi-core architectures. In contrast, both BQ and VBQ scale efficiently with increasing thread counts, maintaining near-linear throughput improvements.

Our composite score (Equation 4), which balances processing speed and storage efficiency, ranks VBQ and BQ as optimal formats with scores of 0.0131 and 0.0170 respectively, an order of magnitude better than other formats, with NAF (0.2440 for FASTA, 0.2435 for FASTQ) also showing competitive combined performance. While NAF excels in storage efficiency, BINSEQ formats maintain superior processing speed, particularly for parallel workloads.

Paired-end results mirror and amplify these trends (Table 3). Multi-threaded BQ achieves 121.71x speedup over FASTQ.GZ and multi-threaded VBQ achieves 43.50x, both substantially higher than their single-end counterparts. This

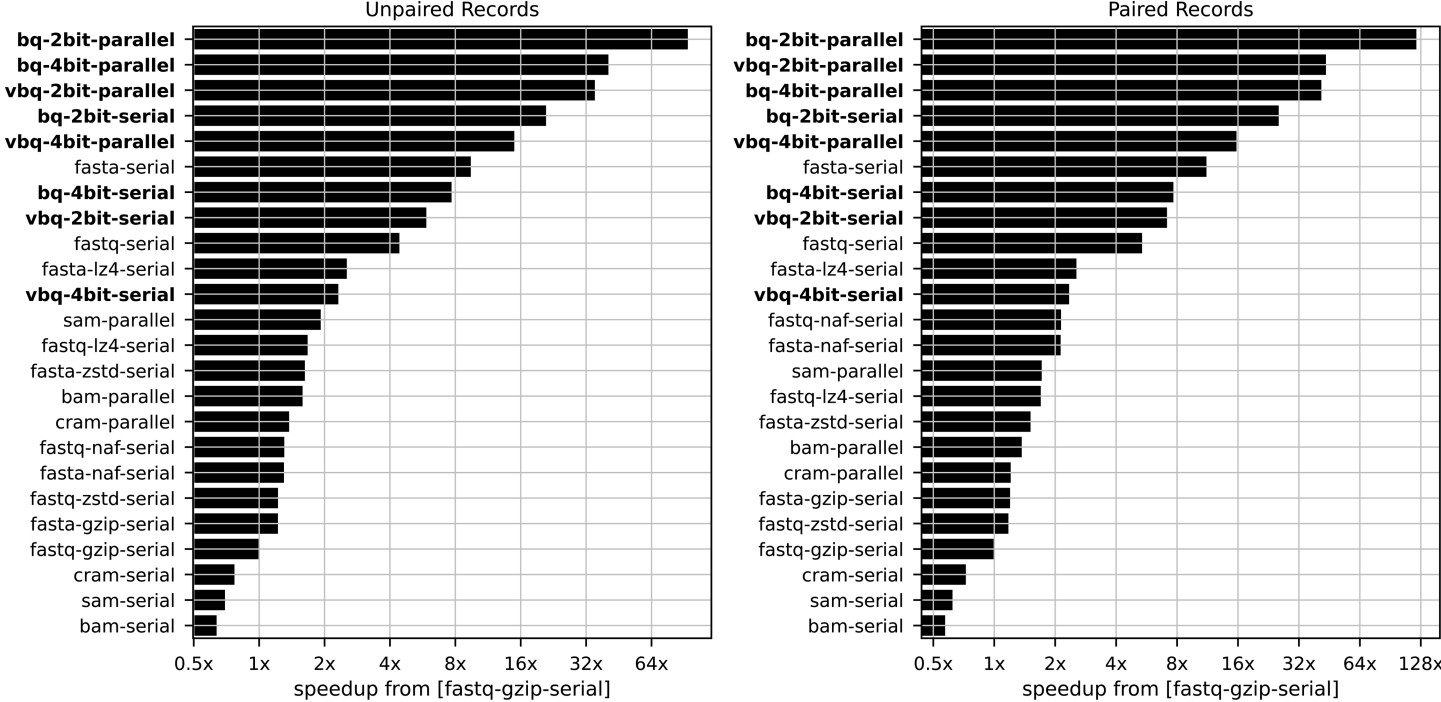

**Fig 2. Speedup of file formats over FASTQ.** GZ for unpaired (a) and paired (b) records. The speedup is measured as the ratio of the total elapsed time to process the record set between the test and reference condition (fastq.gz). Parallel implementations are run using 8 threads.

is consistent with the expectation that processing two records per pair exposes more opportunity for parallelism while the FASTQ.GZ baseline scales poorly with record volume. Notably, for paired-end data BQ achieves the best composite score (0.0071), with VBQ close behind (0.0150), a reversal from the single-end rankings.

Additional analysis of compression algorithms indicates that zstd compression achieves similar compression ratios to gzip but with significantly faster decompression rates for both FASTA and FASTQ. This finding provides additional evidence supporting the replacement of gzip as the de facto compression algorithm in bioinformatics pipelines.

### 3.2  Real data evaluation

To validate our synthetic benchmarks on real sequencing data, we evaluated BQ and VBQ in lossless mode (four-bit encoding with quality scores and sequence headers) across three representative public datasets (Table 4). VBQ achieved consistent 40–47x speedups over gzip-compressed FASTQ at 32 threads across all three data types, while BQ achieved 168–221x speedups on the two fixed-length datasets. Single-threaded performance improvements were more modest (1.8–1.9x for VBQ, 16–17x for BQ), isolating the format-level efficiency gains from the parallelization advantages. Storage requirements for VBQ were 85–88% of the corresponding gzip-compressed FASTQ files, while BQ achieved 54–55% for fixed-length datasets.

### 3.3  K-mer counting

The BINSEQ formats deliver substantial performance advantages for k-mer counting, a fundamental operation in many bioinformatics algorithms. As illustrated in Fig 3a, both BQ and VBQ maintain near-linear scaling of record throughput as thread count increases, continuing to improve even at 128 threads. In stark contrast, FASTQ reaches performance

**Table 3.** Comparison of different compression methods and their performance metrics across various file formats (paired-end records). Time is reported in seconds, size in megabytes. Score is a combined metric of normalized time and size (Equation 4). The "Bit Size" column indicates the number of bits used to represent each nucleotide in the format. The "Parallel" column indicates whether the method was performed in parallel (8 threads). The "Bandwidth" column indicates the Giga-bases per second (Gbp/s). The "Speedup" column indicates the runtime speedup over FASTQ.GZ.

| Format | Variant | Bit Size | Parallel (t=8) | Time (s) | Storage (MB) | Score | Bandwidth (Gbp/s) | Speedup |
|--------|---------|----------|----------------|----------|--------------|-------|-------------------|---------|
| BINSEQ | bq | two | parallel | **0.0382** | 534.06 | **0.0071** | 26.211 | **121.71x** |
| BINSEQ | vbq | two | parallel | 0.1068 | 563.27 | 0.0150 | 9.367 | 43.50x |
| BINSEQ | bq | two | serial | 0.1828 | 534.06 | 0.0161 | 5.472 | 25.41x |
| BINSEQ | vbq | four | parallel | 0.2954 | 603.07 | 0.0318 | 3.386 | 15.72x |
| BINSEQ | vbq | two | serial | 0.6509 | 563.27 | 0.0488 | 1.536 | 7.13x |
| BINSEQ | bq | four | parallel | 0.1125 | 1068.12 | 0.0795 | 8.890 | 41.28x |
| BINSEQ | bq | four | serial | 0.6053 | 1068.12 | 0.1101 | 1.652 | 7.67x |
| FASTQ | naf | eight | serial | 2.1731 | 484.95 | 0.1331 | 0.460 | 2.14x |
| FASTA | naf | eight | serial | 2.1827 | 484.89 | 0.1336 | 0.458 | 2.13x |
| BINSEQ | vbq | four | serial | 1.9815 | 603.07 | 0.1362 | 0.505 | 2.34x |
| FASTA | lz4 | eight | serial | 1.8236 | 1176.19 | 0.1993 | 0.548 | 2.55x |
| FASTA | zstd | eight | serial | 3.0730 | 696.92 | 0.2071 | 0.325 | 1.51x |
| HTSlib | bam | four | parallel | 3.3978 | 611.68 | 0.2219 | 0.294 | 1.37x |
| HTSlib | cram | four | parallel | 3.8526 | **477.98** | 0.2348 | 0.260 | 1.21x |
| FASTA | uncompressed | eight | serial | 0.4155 | 2172.26 | 0.2386 | 2.407 | 11.18x |
| FASTQ | lz4 | eight | serial | 2.7353 | 1198.29 | 0.2577 | 0.366 | 1.70x |
| FASTA | gzip | eight | serial | 3.8793 | 647.89 | 0.2600 | 0.258 | 1.20x |
| FASTQ | zstd | eight | serial | 3.9541 | 735.61 | 0.2736 | 0.253 | 1.17x |
| FASTQ | gzip | eight | serial | 4.6436 | 692.46 | 0.3124 | 0.215 | 1.00x |
| HTSlib | cram | four | serial | 6.4170 | 477.98 | 0.3969 | 0.156 | 0.72x |
| FASTQ | uncompressed | eight | serial | 0.8642 | 4136.83 | 0.5159 | 1.157 | 5.37x |
| HTSlib | bam | four | serial | 8.1194 | 611.68 | 0.5170 | 0.123 | 0.57x |
| HTSlib | sam | eight | parallel | 2.7068 | 4413.39 | 0.6657 | 0.369 | 1.72x |
| HTSlib | sam | eight | serial | 7.4673 | 4413.39 | 0.9606 | 0.134 | 0.62x |

**Table 4.** Real data file size and access speed comparison. VBQ files were encoded in lossless mode (four-bit encoding with quality scores and sequence headers). BQ files use two-bit encoding. Size ratio is relative to FASTQ.GZ. Speedup is the ratio of FASTQ.GZ access time to BINSEQ access time. Parallel benchmarks used 32 threads. All benchmarks were performed on a Dual Intel(R) Xeon(R) Platinum 8468 CPU with NVMe solid-state drives.

| Dataset | Format | Size (GB) | Size Ratio | Time 1T (s) | Time 32T (s) | Speedup (1T / 32T) |
|---------|--------|-----------|------------|-------------|--------------|--------------------|
| scRNA-seq | FASTQ.GZ | 13.44 | 1.00 | 108.4 | — | — |
| (SRR11214033) | BQ | 7.20 | 0.54 | 6.6 | 0.64 | 16.5x / 168.2x |
| | VBQ | 11.43 | 0.85 | 61.7 | 2.69 | 1.8x / 40.3x |
| Short-read WGS | FASTQ.GZ | 49.82 | 1.00 | 393.9 | — | — |
| (SRR36606595) | BQ | 27.41 | 0.55 | 22.6 | 1.79 | 17.5x / 220.6x |
| | VBQ | 43.82 | 0.88 | 208.3 | 8.44 | 1.9x / 46.7x |
| Long-read DNA | FASTQ.GZ | 31.73 | 1.00 | 246.4 | — | — |
| (SRR37455459) | VBQ | 28.08 | 0.88 | 131.3 | 5.21 | 1.9x / 47.3x |

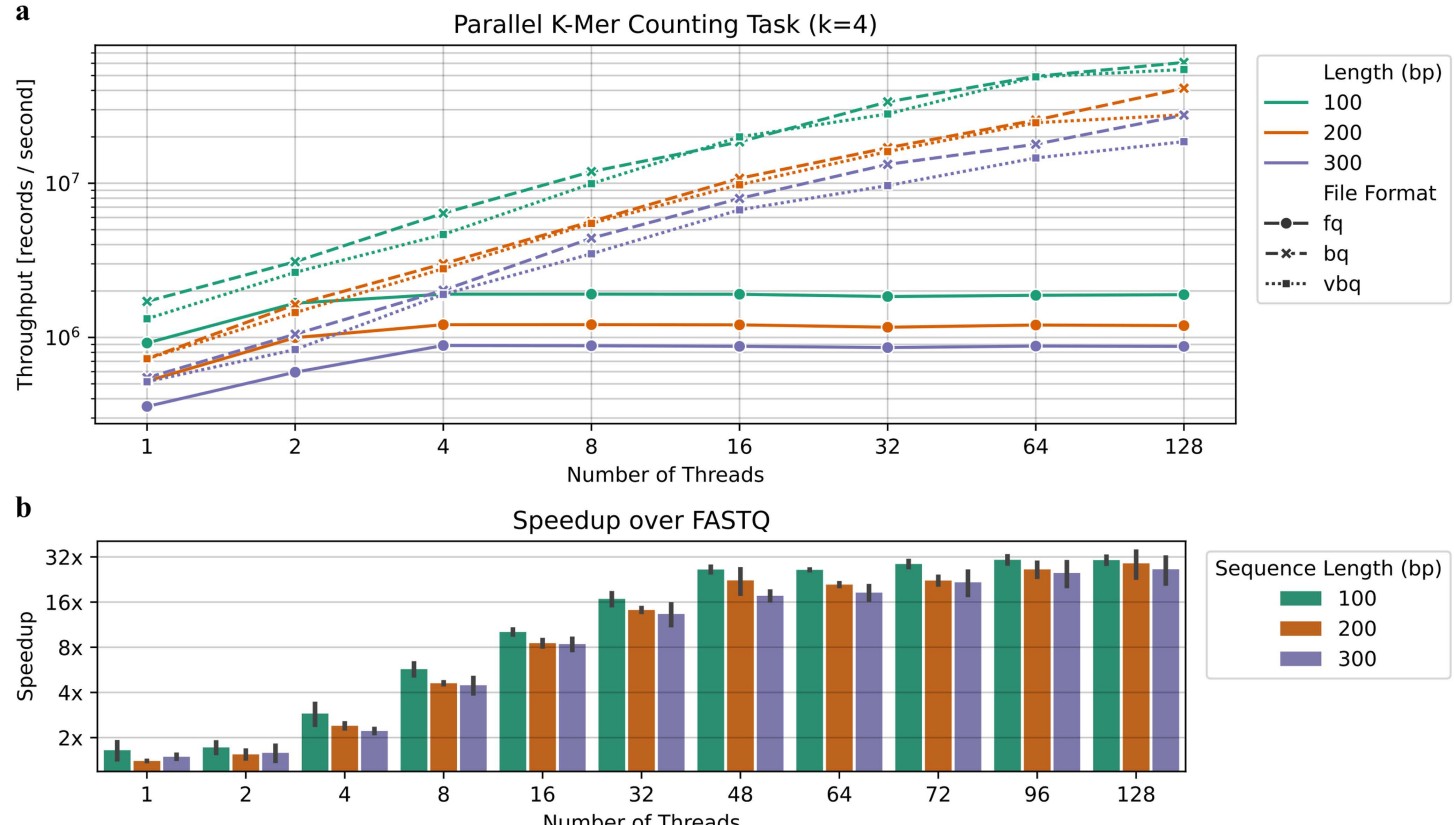

**Fig 3. Runtime characteristics of kmer-count as a function of the number of threads and format: FASTQ (fq), BQ (bq), VBQ (vbq). (a)** Record processing throughput as a function of the number of threads used. **(b)** Speedup (measured as the ratio of the record throughput per second) of BINSEQ over FASTQ as a function of number of threads.

saturation at just 2–4 threads, depending on sequence length, gaining no additional throughput despite allocated computational resources.

This performance differential becomes more pronounced at higher thread counts. By 128 threads, BINSEQ formats achieve near 32x greater throughput per record compared to FASTQ (Fig 3b). This remarkable improvement demonstrates how BINSEQ's architecture effectively overcomes the I/O bottlenecks that constrain traditional formats, allowing full utilization of modern parallel computing resources.

The impact of sequence length on performance scaling is particularly noteworthy. While all formats show some performance reduction with longer sequences, the BINSEQ family maintains its scaling advantage across all tested sequence lengths (100, 200, and 300 bp). This consistent behavior indicates that the performance benefits of BINSEQ formats are robust across varying data characteristics.

K-mer counting, while computationally straightforward, represents a common low-complexity task that forms the foundation of many bioinformatics tools, including de Bruijn graph assemblers, taxonomic classifiers, and sequence similarity analyzers. The demonstrated performance improvements suggest that BINSEQ formats would be particularly beneficial for these applications, especially in high-throughput environments where processing efficiency directly impacts analysis turnaround time.

The ability of BINSEQ formats to maintain scaling efficiency at high thread counts indicates that they effectively shift the performance bottleneck from I/O operations to computational resources, allowing bioinformatics applications to fully leverage available CPU cores. This characteristic is especially valuable in shared computing environments and cloud platforms where computational resources must be efficiently utilized.

## 3.4 Alignment

Sequence alignment represents a more computationally intensive operation than previous benchmarks, yet VBQ still demonstrates significant performance advantages over FASTQ at higher thread counts (Fig 4). This performance difference becomes more pronounced as parallelization increases, showing that BINSEQ formats maintain their advantages even in complex analytical workflows.

Our analysis reveals that the performance divergence point between VBQ and FASTQ depends on both the specific alignment task and the average read length. For short-read whole-genome sequencing, FASTQ and VBQ show comparable performance up to 32 threads across all tested read lengths (Fig 4a), after which VBQ begins to pull ahead. In contrast, for long-read whole-genome sequencing, the performance separation occurs earlier, at approximately 16 threads for all tested read lengths (Fig 4b).

The most striking results appear in the long-read splicing alignment task, where separation points vary with sequence length (Fig 4c). For reads averaging 1000 bp, performance diverges at 16 threads, while for 9000 bp reads, the separation

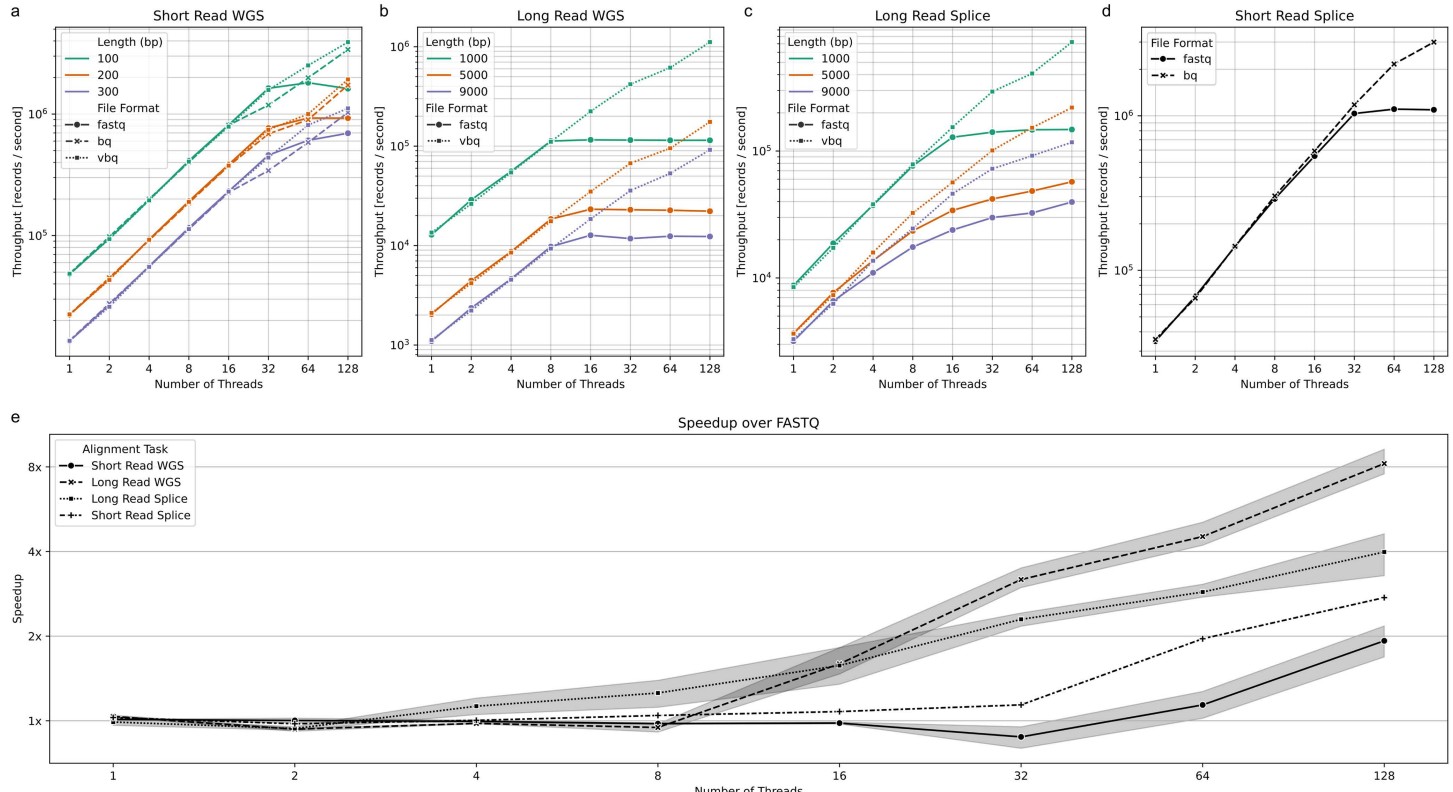

**Fig 4. Runtime characteristics of mmr (a-c) and STAR (d) as a function of the number of threads, format (FASTQ or VBQ), and alignment task. (a)** Short read whole genome sequencing (preset = "sr") **(b)** Long read whole genome sequencing (preset = "map-pb") **(c)** Long read splicing (preset = "splice") **(d)** Short read splicing (STAR) **(e)** Speedup of VBQ or BQ over FASTQ at each thread count for each of the alignment tasks.

occurs as early as 4 threads. Counterintuitively, this complex spliced alignment task shows that VBQ's advantage becomes more pronounced with longer read lengths, where one might expect computational demands to overshadow I/O limitations. This pattern suggests that VBQ's performance advantage stems not merely from more efficient thread utilization but from fundamentally faster record loading to worker threads. Even in this computation-heavy task with 9000 bp reads, FASTQ fails to achieve thread saturation at 128 threads, indicating that the format itself, rather than computational resources, remains a limiting factor.

We also observe a significant divergence in performance between BQ and FASTQ formats for short-read spliced alignments using STAR, where BQ achieves greater than 2x speedup over FASTQ at 128 threads (Fig 4d). This is significant in the case of STAR where the core mapping algorithm was built around streaming records and remains fairly unchanged in this context, so even greater gains can be expected in the future.

The relative speedup of VBQ over FASTQ varies across alignment tasks (Fig 4e), with the most substantial improvements observed in short-read whole-genome and long-read whole-genome sequencing at higher thread counts. All tested scenarios show increasing performance advantages as thread count rises, with no observed plateau for VBQ even at 128 threads.

These results demonstrate that VBQ's benefits extend beyond simple I/O-bound tasks to complex analytical workflows central to genomic research. The format's ability to maintain scaling efficiency with increasing parallelization makes it particularly valuable for computationally intensive applications running on high-performance computing infrastructure.

## 4 Discussion

The development and evaluation of the BINSEQ family of formats demonstrates that significant performance improvements in sequence data processing can be achieved through careful consideration of modern genomics workflows and their computational requirements. By focusing on the core requirements of many bioinformatics applications—efficient access to nucleotide sequences—BQ and VBQ achieve substantial gains in both processing speed and storage efficiency.

The simplicity of BQ's design is one of its key strengths. The format specification is concise and straightforward, consisting of a minimal header structure and a uniform record format that leverages established bit-encoding techniques. This simplicity not only makes the format easy to implement and validate but also contributes to its robust performance characteristics. The use of fixed-size records and two-bit or four-bit nucleotide encodings provide inherent advantages that manifest across various usage scenarios, from basic sequence access to complex analytical workflows. Its definition allows for truly random access and enables tools that were previously bound to a single thread to fully utilize modern multi-core processors.

VBQ, while more complex in implementation, balances flexibility with performance. Although its block-based approach sacrifices the true random access capability of BQ, it compensates by supporting variable-length sequences, quality scores, and optional compression, all while maintaining impressive parallel processing performance. Compared to the de facto standard of gzip-compressed FASTQ, VBQ shows significant improvements with minimal data loss, primarily omitting only sequence headers and invalid nucleotides.

Our performance evaluations demonstrate that both formats translate their design advantages into substantial practical benefits. They consistently outperform traditional approaches across different operational contexts, showing particular advantages in parallel processing scenarios. The ability to scale effectively to high thread counts - achieving up to 94x improvement over compressed FASTQ for sequence access and up to 32x improvement for k-mer based operations - represents a significant advancement for high-throughput genomics applications. This scalability is particularly relevant as the field continues to move toward larger datasets, more computationally intensive analyses, and increased computational availability.

The parallelization advantages of BINSEQ formats operate at the file format level and complement other common parallelization strategies. Large experiments distributed across multiple files can employ file-level parallelism (processing

each file independently), while BINSEQ enables efficient within-file parallelism regardless of data organization. These approaches are composable: workflows can process multiple BINSEQ files in parallel while each file is internally multi-threaded, maximizing utilization of available computational resources across diverse computing environments.

It is worth noting, however, that the magnitude of within-file gains depends on the degree to which I/O is the limiting factor in a given workflow. When a large experiment is distributed across many independent compressed FASTQ files and each file is processed by a separate worker, a sufficiently large number of concurrent readers can collectively saturate the throughput of the underlying storage medium regardless of file format, reducing BINSEQ's advantage relative to the single-file case. In practice, not all experiments are large enough to be split across many files, not all tools are designed to consume multiple files in parallel, and achieving full storage saturation with gzip-compressed FASTQ requires substantially more concurrent processes than with BINSEQ. We therefore expect that in common workflows, where computations are embarrassingly parallel but data are organized into a modest number of files, BINSEQ-enabled pipelines will deliver considerable speedups, while in workflows where storage I/O is already saturated, the gains will be more modest but BINSEQ still offers reduced CPU overhead and simpler parallelization at the library level.

We also note the existence of tools that parallelize gzip decompression itself, most notably pugz [9] and rapidgzip [10], which employ speculative or block-caching strategies to decompress gzip streams using multiple threads. Unlike standard multi-threaded compressors such as pigz, these tools support parallel decompression but at the cost of substantially increased CPU usage per byte decompressed. We did not include them in our benchmarks as they are not currently integrated into standard bioinformatics pipelines or sequence parsing libraries, and thus do not reflect the decompression performance that users encounter in practice. BINSEQ's architecture avoids the need for speculative computation entirely, providing inherently parallel access through its block-based (VBQ) or fixed-record (BQ) structure with native random access that can be directly leveraged by downstream tools through the BINSEQ library API.

Importantly, we found that the performance advantages of BINSEQ formats extend beyond simple data retrieval tasks. Even in computationally intensive operations like sequence alignment, where the processing time per record is substantially higher, both formats demonstrated improved thread utilization and record throughput compared to traditional formats. This suggests that the bottleneck created by sequential parsing and decompression in FASTQ affects not just I/O-bound applications but also more complex analytical workflows.

The integration of BINSEQ formats with existing tools, demonstrated through our adaptation of minimap2, shows that the formats can be readily incorporated into established bioinformatics pipelines. The provided library and command-line tools offer a framework that simplifies both the adoption of BINSEQ formats in existing applications and the development of new tools that can take full advantage of their performance characteristics. The hook-based parallel processing interface, in particular, provides a flexible foundation for building high-performance genomic analysis tools.

For practitioners, the choice between BQ and VBQ presents a clear trade-off between performance and flexibility. BQ is optimized for maximum throughput and minimal storage with fixed-length sequencing data, making it ideal for platforms like Illumina that produce standardized reads. VBQ offers greater versatility for variable-length sequences and applications where quality scores remain important, such as long-read sequencing from PacBio or Oxford Nanopore technologies, with only a modest performance decrease compared to BQ but still substantial improvements over traditional formats.

The trade-offs inherent in the BINSEQ family's design reflect a broader consideration in bioinformatics tool development: the balance between generality and optimization for common use cases. While FASTQ's flexibility has contributed to its widespread adoption, our results suggest that there is significant value in optimizing for the specific requirements of modern high-throughput sequencing applications. One important consideration is the handling of ambiguous nucleotides (N bases), which we address through configurable policies (skip, random replacement, fixed replacement, or preservation via four-bit encoding) rather than prescribing a single approach. The optimal policy is application-dependent—ambiguous bases in single-cell barcodes may require different treatment than those in genomic reads for variant calling. This same

consideration applies to quality scores, which can be optionally preserved or discarded depending on the specific needs of the application. While this consideration applies to all sequence formats, BINSEQ provides explicit policy choices at the format level while maintaining high performance regardless of policy selected. The performance improvements demonstrated by BQ and VBQ indicate that the bioinformatics community might benefit from reconsidering the one-size-fits-all approach to sequence data storage, particularly as dataset sizes continue to grow and computational efficiency becomes increasingly critical.

Looking forward, the BINSEQ architecture provides a foundation for further optimization and extension. The reserved space in both formats' headers and the flexible flag fields offer clear paths for adding features while maintaining backward compatibility. Future development could explore additional compression schemes, integration with cloud-native storage, handling of higher-order segments per record, and structured data provenance, all while preserving the core performance benefits of the current implementation.

In conclusion, the BINSEQ family represents a significant step forward in sequence data storage and processing efficiency. Their combination of simplicity, performance, and practicality makes them valuable additions to the bioinformatics toolkit, particularly for applications where computational efficiency is paramount. While not complete replacements for existing formats in all scenarios, BQ and VBQ demonstrate that substantial improvements in processing efficiency are achievable through careful consideration of modern genomics workflows and their computational requirements.

## Supporting information

**S1 Table. Nucleotide encoding.** Standard nucleotides use two-bit encoding. In four-bit mode, these values are preserved in the least significant bits with leading zeros, while N requires the full four-bit representation.
(PDF)

**S2 Table. BQ header structure.** Complete specification of the 32-byte BQ header including field offsets, sizes, types, and descriptions for magic number, version, sequence lengths, bit encoding, flags, and reserved bytes.
(PDF)

**S3 Table. VBQ header structure.** Complete specification of the 32-byte VBQ header including format version, virtual block size, and boolean flags indicating presence of quality scores, compression, paired-end reads, sequence headers, and record flags.
(PDF)

**S4 Table. VBQ block header.** Specification of the 32-byte block header structure including magic number, true block size (accounting for compression), number of records in block, and reserved bytes for future extensions.
(PDF)

**S5 Table. VBQ record structure.** Detailed field-level specification of variable-length VBQ records including flags, sequence lengths, encoded nucleotide buffers, quality scores, and sequence headers for both primary and extended (paired-end) sequences.
(PDF)

**S6 Table. VBQ index header.** Specification of the 32-byte index header structure including magic number, file size verification field, and reserved bytes for future extensions.
(PDF)

**S7 Table. VBQ index range.** Specification of the 32-byte index range structure describing block locations, including file offset, block size, number of records per block, cumulative record count, and reserved bytes.
(PDF)

**S8 Table. Parallel processing hooks.** Description of the map-reduce interface for parallel processing including the process_record map function for individual record processing and the on_batch_complete reduce function for batch aggregation.
(PDF)

## Acknowledgments

The authors would like to thank David Holtz for his help on improving BINSEQ and its ecosystem, as well as Hani Goodarzi, Nicholas D. Youngblut, Yusuf H. Roohani, Abhinav Adduri, David P. Burke, and the rest of the Arc Institute computational team for their helpful suggestions and feedback.

## Author contributions

**Conceptualization:** Noam Teyssier.

**Data curation:** Noam Teyssier.

**Formal analysis:** Noam Teyssier.

**Investigation:** Noam Teyssier.

**Methodology:** Noam Teyssier, Alexander Dobin.

**Project administration:** Noam Teyssier.

**Software:** Noam Teyssier.

**Supervision:** Alexander Dobin.

**Validation:** Noam Teyssier.

**Visualization:** Noam Teyssier.

**Writing – original draft:** Noam Teyssier.

**Writing – review & editing:** Noam Teyssier.

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
