## [Decision Letter · Decision Letter 0]

5 Aug 2025

PCOMPBIOL-D-25-01020

BINSEQ: A Family of High-Performance Binary Formats for Nucleotide Sequences

PLOS Computational Biology

Dear Dr. Teyssier,

Thank you for submitting your manuscript to PLOS Computational Biology. After careful consideration, we feel that it has merit but does not fully meet PLOS Computational Biology's publication criteria as it currently stands. Therefore, we invite you to submit a revised version of the manuscript that addresses the points raised during the review process.

Please submit your revised manuscript within 60 days Oct 05 2025 11:59PM. If you will need more time than this to complete your revisions, please reply to this message or contact the journal office at ploscompbiol@plos.org. Please include the following items when submitting your revised manuscript:

We look forward to receiving your revised manuscript.

Kind regards,

Stephan Schiffels

Academic Editor

PLOS Computational Biology

Ilya Ioshikhes

Section Editor

PLOS Computational Biology

**Additional Editor Comments:**

All three reviewers found your paper of high relevance, being well written, and in principle suitable for publication in this journal after a major revision.

While all reviewers appreciated the significant increase in I/O access perfomance that the new formats provide, there were shared concerns on some of the design choices which may limit the usability of the new formats. In particular, Reviewers 1 and 2 found the choice to drop quality scores from the BINSEQ format to be rather case-limiting. Both reviewers also criticized the current lack of read names in the formats. All reviewers mentioned the problem of 2-bit encoding of nucleotides, which excludes missing bases ("N") or ambiguous base calls.

There are also many more detailed issues raised by the reviewers that I hope you find helpful to revise your manuscript, which I look forward to receive.

**Journal Requirements:**

4) We notice that your supplementary Tables are included in the manuscript file. Please remove them and upload them with the file type 'Supporting Information'. Please ensure that each Supporting Information file has a legend listed in the manuscript after the references list.

**Reviewers' comments:**

Reviewer's Responses to Questions

**Comments to the Authors:**

Reviewer #1: Teyssier and Dobin described BINSEQ and VBINSEQ, two binary formats for storing NGS reads with I/O performace being a primary focus. BINSEQ is designed for reads of fixed length while VBINSEQ for reads of variable length and optionally keeps base quality. Neither format stores read names. The format design is straightforward and the implementation is well engineered. The authors BINSEQ greatly outperforms other formats and associated tools. Critical comments are as follows:

1) As a general comment, I see BINSEQ will have limited use cases. Base quality is essential to variant calling. This is why Illumina tried to reduce quality resolution but still decided to use 2 bits per quality. Base quality is unimportant to RNA-seq quantification, but read lengths could be variable due to trimming. BINSEQ wouldn't work in this case, either.

2) As the more flexible format, VBINSEQ should optionally store read names. Illumina read names carry information and are still used for marking optical duplicates if I am right. Probably removing names will have little impact to analysis but if the authors want VBINSEQ to be an archival format, a lossless mode would be essential to data centers. The authors can use the flag field to indicate the presence/absence of read names.

3) How are non-A/C/G/T bases stored? The UCSC 2-bit format uses a sparse vector to keep positions of non-A/C/G/T bases. The authors can do the same for sequences with "N" and use a flag bit to indicate the presence/absence of this information.

4) Zstd and pigz (a popular gzip alternative) support multi-threading. Please evaluate their multi-threading mode in Table 2. By the way, on the Time column, better to use 0.21, 0.05, 0.86, etc for timing. This is easier to read in my opinion.

5) What data is used for benchmarking? Please specify the resolution of base quality, the format of read names, the average read depth and whether input reads are sorted by coordinate. Some of these factors may greatly affect compression ratio.

Reviewer #2: In this manuscript, the authors present a family of two binary file formats for nucleotide sequences. First, the BINSEQ format, which is designed for storing sequences of a fixed length. Second, the VBINSEQ format, an extension to BINSEQ that can contain sequences of variable length. With the development of these formats, the authors seek to increase the speed at which sequences can be accessed, and hence the speed of computations reading that data. To showcase this improvement, the authors compare the speed and storage footprint of their formats to other standard formats, and particularly the FASTQ/FASTA formats, which currently serve as the de facto default formats for the storage, sharing, and archiving of sequence data.

Overall, I feel that the authors created suitable and fair tests to assess the efficacy of the BINSEQ formats. Furthermore their tests showcase that these formats achieve their goal of increasing the parallelisation efficiency of operations, while maintaining a storage footprint similar in size to FASTQ files.

While this is not the first attempt to improve on sequence data formats, it is the first approach I have seen to focus on improving the access speed of the data, instead of only their storage footprint. With the ever-increasing volume of sequencing data being produced, storing this data in optimised data formats that are interoperable and easily convertible is more relevant than ever.

However, there are design decisions made in the creation of these formats that might severely limit their applicability across the genomics field. Namely, the current omission of quality scores, sequence identifiers, and uncertain base calls and/or Ns in the stored sequences. Admittedly, my own experience is limited to low-coverage ILLUMINA data from human DNA, so I acknowledge that perhaps the omitted information is less important in when working with other types of sequencing data.

Should changes to this format allow for the inclusion of the omitted information (thus making it fully equivalent to FASTQ), BINSEQ would become a valuable data format for genomics in the age of HPCs and cloud computing.

Major comments:

1. It is not entirely clear what niche the BINSEQ format currently fills. The lack of any quality scores makes it suboptimal for storing raw reads (like a FASTQ does), while the specification that sequences must be of equal length makes it unsuitable for storing reference sequences composing multiple contigs (like a FASTA does).

2. Considering that this file format is designed for storing reads before any alignment, the decision to not allow for Ns in the nucleotide sequences in either format is dangerous when working with lower-coverage data. This becomes a bigger issue in the BINSEQ format, where quality scores are not accounted for.

3. More broadly, I am very skeptical of the decision to drop quality scores from the BINSEQ format specification altogether. The authors argue that "quality scores and sequence identifiers [are] frequently ignored by modern bioinformatics tools", and hence can be dropped from BINSEQ files entirely. This might be okay to do if everything goes according to plan during data generation, but it allows for no troublshooting should anything go wrong (e.g. if some part of a flowcell produces poor quality data).

Minor comments:

1. Please ensure the legends of tables and figures are accurately describing the content. For example:

a. Table2: The legend states the times given are in milliseconds, but the unit stated is (s).

b. Figure 4e: The legend states this plot shows the speedup of VBINSEQ, but the figure includes results for short-read spliced alignment, which was only ran with BINSEQ, and not VBINSEQ.

Reviewer #3: In this manuscript, Teyssier and Dobin introduce the BINSEQ format (both the simple binseq and more flexible vbinseq) formats. They describe the format in detail, argue for its utility compared to the FASTQ format for storing primary sequencing data; especially in the context of high-throughput analysis tasks such as k-mer counting and read alignment via case studies with modified versions of existing tools. They also introduce a suite of tools for handling and converting to and from these formats. Overall, I share the concerns of the authors about the fundamental way in which the FASTQ format is ill-suited for modern parallel processing tasks, and shudder to think how much time (both CPU time and wall-clock time) has been wasted simply parsing a format that was never well-designed for the purposes of parsing by a computer. The BINSEQ format is a big step up from FASTQ but ultimately, as the authors themselves suggest, whether or not a true alternative can emerge at scale will depend on ecosystem an tool adoption. Nonetheless, it is certainly a worthwile goal. The manuscript itself is concise and well-written. I have several questions and suggestions for the authors.

Questions / suggestions:

* One key reference that is missing from the current manuscript is "Scaling read aligners to hundreds of threads on general-purpose processors"[^1], by Langmead et al. It is directly relevant to the current manuscript and describes precisely the scalability bottleneck upstream of multi-core read alignment that arises from the fundamentally unparallelizable nature of parsing a (potentially compressed) FASTQ file.

* A question that arises to me in the format is the extent to which assuming that sequence records have either 1 or 2 segments may be problematic. Certain protocols, though relatively rare, make use of more than 2 segments per record to encode information (e.g. some single-cell ATAC-seq protocols have 3 segments per record, with the first 2 representing a biological paried-end read, and the third representing a technical barcode). The authors should consider if and how this might be supported in binseq (presumably vbinseq) or justify excluding support. It seems to me it would be fairly easy to have the number of segments per records be a simple field recorded in the header, and that having values larger than 2 doesn't seem to pose any fundamental tension with the rest of the format (as long as it is fixed throughout the whole file).

* Related to the above, what replacement policy was used in the experiments conducted in the paper?

* In general, I think a somewhat deeper exploration of the effect of different ambiguous base policies should be conducted, or an alternative ambiguous base policy should be included. Specifically, the current approach can skip records with ambiguous bases, replace those bases with pseudorandom nucleotides, or replace those bases with a fixed nucleotide. These are all reasonable policies. However, I feel it is important to characterize the effect of such a policy on downstream analysis. For example, if an `N` nucleotide appears in the barcode of a single-cell RNA-seq read, then downstream algorithms may handle this differently than if a fixed nucleotide is present (in fact, by replacing the `N` with a fixed nucleotide, a barcode collision may be artificially introduced). Further, in other analyses, tools may have special ways to handle ambiguous bases (assigning them a different cost during alignment). I suspect that in the vast majority of cases, the policies supported by binseq will have no effect on the subsequent analysis, but that may not always be the case (e.g. variant detection from fairly low-coverage data). This is worth further exploration in the manuscript in my opinion. Likewise, perhaps it makes sense to offer a policy where the ambiguous bases are recorded (either as auxiliary data - a list of positions for a record - or in an alternative encoding that allows more than 4 characters). The same comment also applies to the skipping of quality scores in the basic binseq format.

* Another question that occurred to me is if it is the case that in all experiments in the paper, where speeds are compared over different formats, if the experimental samples processed all consist of individual files or file pairs. That is, for large samples, the input is often spread across multiple files (e.g. a large single-cell RNA-seq experiment may have `sample1_lane1_read1_001`, `sample1_lane1_read_2_001`, `sample1_lane1_read1_002`, `sample1_lane1_read2_002`, ..., etc.). In such cases, it is often possible to parse fastq input truly in parallel, at least at the scale of the number of input file streams. While this is certainly (a) not ubiquitous and (b) not taken advantage of by all tools, I think it would be very useful to the reader to know if any of the samples processed in the benchmarks exhibited this quality, and if not, to perhaps include some. This will give a more comprehensive overview of the expected throughput benefits of binseq.

* Finally, I have a specific design suggestion, which the authors may of course feel free to accept or not. While having an index for random block-level access in vbinseq mirrors the way that e.g. SAM/BAM indices work, I actually feel this is a suboptimal design. These indices tend to be tiny and fast to create, which is why we tend to re-create them on demand. However, it does constitute extra work. Further, such indices are separate files and easy to forget or misplace when transferring data (e.g. one could easily imagine a use grepping for `.vbq` files and forgetting to include all of the index files when creating an archive or copying files between machines). Since the indices are only relevant in the context when the entire file exists on disk (i.e. not in the streaming context), I would recommend the authors consider including the index for vbinseq files as a footer.

Here is the proposal: the vbinseq format can allow an optional footer. This footer comes after all of the record blocks. If this footer is present, the file ends in a magic number (which validates the existence of the footer). The word before the magic number then stores the size of the index, so that the bytes from (EOF - 8 - size_of_magic_number - index_size) until (EOF - 8 - size_of_magic_number) encode the index itself. This seems as it should work because the index can then be calculated in either a streaming manner as the file is created, which is OK because one won't know the index until all blocks have been compressed, and then appended to the end. Likewise, for a file without the footer, one could easily calculate the index as is currently done and then, rather than write it as a separate file, simply append the footer to the input file. I suspect that this approach may be useful as it keeps the index along with the rest of the data, but does so in a manner that doesn't preclude streaming or parallel writing / appending of the record blocks.

Minor points:

* On page 4, the statement that the encoding and decoding operations are of sub-linear time complexity don't quite seem right. Certainly, the input must be read, which is itself an O(N) operation for an input of length N. Further, even if one is operating a word at a time and achieving N/(W/sigma) work (for W-bit words, a sigma-bit alphabet encoding and a string of N symbols), this is still linear in the length of the input (i.e. it is still O(N), just with a constant c < 1).

* In section 2.2.2 the authors describe the input and file i/o capabilities of their binseq library and the C and C++ bindings. I took a look at the bindings, and I think it is great that the authors provide these. One question that this brought up to me is whether or not binseq is designed to allow streaming input (I think this would be immensely useful). If so, I'd encourage the authors to provide an additional C / C++ function for opening a binseq file for reading apart from `mmap_open`, one that may be properly suited for reading the file in a streaming manner.

* In section 2.3.3 it is stated that minimap2 specifically notes that it is not designed for the use-case of spliced short-read alignment. However, as of release 2.29 this is no longer true[^2]. To be clear, I *don't* think the authors need to add minimap2-based short-read, spliced alignment benchmarks to their paper, but rather they should remove or amend this particular justification. Since STAR is, by far, one of the most common short-read RNA-seq spliced read aligners, I think it is fine to having testing for this datatype using only that tool

Finally, this may be out of scope for this paper which defines and justifies (well, in my opinion) the BINSEQ format. However, having a new format and generating a lot of new data to place into this format, seems to me an opportunity to improve the state of data provenance tracking and automated metadata propagation. Might the authors consider if it makes sense to have a structured metadata section in the binseq file that can record relevant information about the source of the data (e.g. sequencing data, machine, etc.)? Perhaps the relevant full set of information is not something that the authors feel comfortable specifying unilaterally, which I think makes sense. However, having the format have built-in support for metadata (even flexible metadata where much of it is optional) could, itself, have

References

----------

[^1]: Ben Langmead, Christopher Wilks, Valentin Antonescu, Rone Charles, Scaling read aligners to hundreds of threads on general-purpose processors, Bioinformatics, Volume 35, Issue 3, February 2019, Pages 421–432, https://doi.org/10.1093/bioinformatics/bty648

[^2]: https://lh3.github.io/2025/04/18/short-rna-seq-read-alignment-with-minimap2

**Have the authors made all data and (if applicable) computational code underlying the findings in their manuscript fully available?**

Reviewer #1: Yes

Reviewer #2: Yes

Reviewer #3: Yes

PLOS authors have the option to publish the peer review history of their article (what does this mean?). If published, this will include your full peer review and any attached files.

Reviewer #1: No

Reviewer #2: No

Reviewer #3: **Yes:** Rob Patro

**Figure resubmission:**
---

## [Decision Letter · Decision Letter 1]

9 Dec 2025

PCOMPBIOL-D-25-01020R1

BINSEQ: A Family of High-Performance Binary Formats for Nucleotide Sequences

PLOS Computational Biology

Dear Dr. Teyssier,

Thank you for submitting your manuscript to PLOS Computational Biology. We see that the manuscript has substantially improved, but still requires a minor revision in order to fully meet PLOS Computational Biology's publication criteria. Therefore, we invite you to submit a revised version of the manuscript that addresses the points raised during the review process.

We look forward to receiving your revised manuscript.

Kind regards,

Stephan Schiffels

Academic Editor

PLOS Computational Biology

Ilya Ioshikhes

Section Editor

PLOS Computational Biology

**Additional Editor Comments:**

You will see that all reviewers agree that the manuscript has substantially improved. Reviewers 1 and 3 ask for minor revisions. Specifically, reviewer 1 suggests one last additional test on real data to be done, but they explicitly suggest a smaller scope than the other tests/benchmarks in the paper. Reviewer 3 suggests additional discussion of trade-offs related to multi-file inputs and I/O saturation. I do follow the two reviewers' arguments here, and therefore also decide for minor revision.

**Journal Requirements:**

We ask that a manuscript source file is provided at Revision. Please upload your manuscript file as a .doc, .docx, .rtf or .tex. If you are providing a .tex file, please upload it under the item type u2018LaTeX Source Fileu2019 and leave your .pdf version as the item type u2018Manuscriptu2019.

**Reviewers' comments:**

Reviewer's Responses to Questions

**Comments to the Authors:**

Reviewer #1: The authors have addressed most of my concerns. I have a few more comments:

1) This is a follow-up to my initial comment 5). I didn't realize that the authors are using simulated data. Please benchmark on real data in the VBQ lossless mode. They do not need to do comprehensive evaluation like Fig. 2. Just gzip/VBQ file sizes and decompression speed would suffice. Ideally, they could include short genomic reads, long genomic reads and short RNA-seq reads.

2) This is a comment on the new results. High-performance tools such as kmc and minimap2 often use separate I/O threads such that we can read/write data while performing the core task. Does mmr support separate I/O threads? If not, it would be good to show how minimap2 works given gzip'd input. It would be good to test on real data in the VBQ lossless mode.

3) [Discretionary] I encourage the authors to look beyond their own environment. In the wider ecosystem, WGS data dominates in terms of data volume and most production teams would choose a lossless format. Note that the CRAM format was published in 2011 but didn't get much attention initially. It was gaining popularity primarily due to its lossless-by-default mode. Samtools/picard still supports lossy options but few are enabling them. This is why I am requesting the evaluation of the VBQ lossless mode, the only mode that matters to those outside the Arc Institute.

Reviewer #2: The changes made to VBQ to optionally include base qualities, read names (--archive), as well as the possibility to use four-bit encoding for Ns have largely addressed my concerns with the applicability of the BQ and VBQ file formats.

I thank the authors for implementing these changes.

Reviewer #3: I thank the authors for their detailed responses to my and the other reviewer's concerns with the initial manuscript. I find the manuscript itself, and the Binseq tooling to be substantially improved. The ability to optionally store 'N' values, quality scores, and read header information increases the broadness of VBQs applicability. I am glad that the suggested idea of a footer-based index for VBQ worked out, and, generally, am very heartened to see all of the progress and improvements made to the software since the initial version of the manuscript. Likewise, I am also encouraged by the authors' perspective on Binseq as a community driven standard and library, and am glad to see that others have, indeed, been contributing to the project. I feel these developments bode well for the future of Binseq.

While I agree with the authors that several of the points brought up in my original review might have expanded the scope of the manuscript more than they wanted in this initial paper describing Binseq, I believe there are two specific points that are worth at least a little bit more discussion.

First, the authors clarify that they focus here on format-specific gains, and hence choose not to compare the throughput obtained by BQ and VBQ to that obtained by other formats in the case of multi-file input (i.e. when an experiment may be broken into many files). I will not press the point, specifically, that a particular comparison needs to be presented in this manuscript (though I still think it might be informative). However, I do think it is worthwhile addressing, at a high-level, where and how this might affect the tradeoff. That is, while it is true that the improved throughput of Binseq can conceptually be *stacked* with the improved performance that comes from handling multiple input files in parallel, it is also true that, for sufficiently large numbers of individual input files, I/O may become saturated anyway. That is, BQ and VBQ are more efficient than compressed fasta/fastq files for numerous reasons, but one substantial reason is that they can be easily chunked and processed in parallel. This allows extracting more throughput from the underlying storage than can be done from a compressed fasta/fastq file. However, given a sufficient number of independent compressed fasta/fastq file, a client program could be fed at the speed of storage throughput. In that scenario, while I'd expect Binseq to be somewhat faster, I expect those gains to be lesser than in the single-file or small number of files case. I don't view this as an argument against Binseq in any way, since, as I specified in my previous review and as the authors mention, this practice is not ubiquitous, not all experiments are large enough to use this approach, and not all tools take advantage of this capability. However, in order to give the reader a fair idea of the spread of potential gains, this would be useful to mention. Generally, I'd expect that, in typical / common workflows where the computations are embarrassingly parallel, Binseq-enabled pipelines would be considerably faster than those based on compressed fasta/fastq. However, in workflows where I/O isn't a bottleneck, or in workflows where I/O can already be saturated with existing fasta/fastq formats, the speedup may be more modest.

Second, reviewer 1 previously mentioned pigz and zstd supporting multithreading. While the authors are correct that these tools support only multi-threaded compression and not multi-threaded decompression, it is worth mentioning that there exist a couple of tools that support the latter; namely, pugz (https://github.com/Piezoid/pugz) and more recently rapidgzip (https://github.com/mxmlnkn/rapidgzip). Both of these approaches are based on speculative decompression and/or caching, and therefore trade-off extra computation in order to achieve parallel decompression. Nonetheless, with sufficient extra computation, they can yield substantial speedups over even the best implementations of serial gzip decompression (i.e. that provided by Intel's isa-l library (https://github.com/intel/isa-l)). I will leave it up to the authors as to how to address the existence of these related libraries. I highly suspect that even the state-of-the-art speculative parallel decoder (rapidgzip) won't be competitive with Binseq, as the amount of extra computation required to obtain much faster decompression is relatively large. However, even if these tools are not included or used in the benchmarks of the paper, they are related to the goal of the paper and likely deserve mention (both pugz : http://www.hicomb.org/HiCOMB2013/papers/HICOMB2019-07.pdf and rapidgzip : https://dl.acm.org/doi/10.1145/3588195.3592992 have corresponding published papers).

**Have the authors made all data and (if applicable) computational code underlying the findings in their manuscript fully available?**

Reviewer #1: Yes

Reviewer #2: Yes

Reviewer #3: Yes

PLOS authors have the option to publish the peer review history of their article (what does this mean?). If published, this will include your full peer review and any attached files.

Reviewer #1: No

Reviewer #2: No

Reviewer #3: **Yes:** Rob Patro

**Figure resubmission:**
---

## [Editor Report · Decision Letter 2]

30 Mar 2026

Dear Dr Teyssier,

We are pleased to inform you that your manuscript 'BINSEQ: A Family of High-Performance Binary Formats for Nucleotide Sequences' has been provisionally accepted for publication in PLOS Computational Biology.

Best regards,

Stephan Schiffels

Academic Editor

PLOS Computational Biology

Ilya Ioshikhes

Section Editor

PLOS Computational Biology

I am pleased with the additional revisions and believe they address the reviewers' remarks. I accept this article for publication.

---

## [Editor Report · Acceptance letter]

PCOMPBIOL-D-25-01020R2

BINSEQ: A Family of High-Performance Binary Formats for Nucleotide Sequences

Dear Dr Teyssier,

I am pleased to inform you that your manuscript has been formally accepted for publication in PLOS Computational Biology. Your manuscript is now with our production department and you will be notified of the publication date in due course.

With kind regards,

Anita Estes
